# Environmental-Friendly Adsorbent Composite Based on Hydroxyapatite/Hydroxypropyl Methyl-Cellulose for Removal of Cationic Dyes from an Aqueous Solution

**DOI:** 10.3390/polym14112147

**Published:** 2022-05-25

**Authors:** Noureddine Akartasse, Khalil Azzaoui, Elmiloud Mejdoubi, Belkheir Hammouti, Lhaj Lahcen Elansari, Mohamed Abou-salama, Mohamed Aaddouz, Rachid Sabbahi, Larbi Rhazi, Mohamed Siaj

**Affiliations:** 1Laboratory of Applied Chemistry and Environment, Department of Chemistry, Faculty of Sciences, Mohammed 1st University, P.O. Box 717, Oujda 60000, Morocco; k.azzaoui@yahoo.com (K.A.); ee.mejdoubi@gmail.com (E.M.); hammoutib@gmail.com (B.H.); elansarihassan@yahoo.fr (L.L.E.); m.aaddouz@ump.ac.ma (M.A.); 2CREHEIO, Ecole des Hautes Etudes d’Ingénierie, Oujda 60000, Morocco; 3Laboratory of Molecular Chemistry, Department of Chemistry, Materials and Environment, Faculty Multidisciplinary Nador, University Mohammed Premier, B.P. 300, Selouane, Nador 62700, Morocco; m.abousalama@ump.ac.ma; 4Higher School of Technology, University of Ibn Zohr, Quartier 25 Mars, P.O. Box 3007, Laayoune 70000, Morocco; r.sabbahi@uiz.ac.ma; 5Laboratory of Plant Biotechnology, Faculty of Sciences, University of Ibn Zohr, Hay Dakhla, P.O. Box 8106, Agadir 80000, Morocco; 6Institut Polytechnique UniLaSalle, Transformations & Agro-Resources Research Unit (ULR7519), 19 rue Pierre Waguet, BP 30313, 60026 Beauvais, France; 7Department of Chemistry and Biochemistry, Université Du Québec à Montréal, Montréal, QC H3C 3P8, Canada

**Keywords:** nanocomposite, hydroxypropyl methylcellulose, hydroxyapatite, methylene blue, adsorption

## Abstract

The aim of this study is to develop a new, efficient, and inexpensive natural-based adsorbent with high efficacy for the cationic dye methylene blue (MB). A natural-based nanocomposite based on hydroxyapatite (HAp) and hydroxypropyl methylcellulose (HPMC) was selected for this purpose. It was synthesized by the dissolution/reprecipitation method. A film with a homogeneous and smooth surface composed of nanoparticles was prepared from the nanocomposite. HPMC and HAp biopolymers were selected due to their compatibility, biodegradability, and non-toxicity. Total reflectance infrared spectroscopy (ATR-FTIR), scanning electron microscopy (SEM), and calorimetric/thermal gravimetric (DSC/TGA) analysis results revealed the existence of strong physical interaction between the composite components. Scanning electron microscopy (SEM) observations show a composite sheet with a homogenous and smooth surface, indicating excellent compatibility between HPMC and HAp in the composite. The nanocomposite was evaluated as an adsorbent for organic dyes in an aqueous solution. The effects of solution pH, initial MB concentration, composite concentration, and adsorption time on the adsorption efficiency were evaluated. The highest adsorption rate was seen as 52.0 mg of MB/g composite. The adsorption rate reached equilibrium in about 20 min. Fitting of the adsorption data to the Langmuir and Freundlich adsorption models was investigated. Results showed that the adsorption process follows the Langmuir isotherm model. The kinetic study results revealed that the adsorption process was pseudo-second-order. The herein composite is an excellent alternative for use as contemporary industrial-scale adsorbents.

## 1. Introduction

Organic dyes from factories such as textiles, cosmetics, gasoline, printing, and plastics generate toxic substances that cause a significant pollution issue when discarded into the waste stream without any further treatment [1]. This is considered an emergent issue since such industries are not paying attention to the treatment of dyes before discharge to the environment. This behavior might have a severe impact on the health of humans, animals, and living organisms [2]. The biological degradation of the generated dyes is not always possible [2] since they are known to have a complex aromatic molecular structure that makes them very stable and display a xenobiotic behavior [3].

So, removal of these pollutants by other means becomes a high priority. Several methods are known to be useful in this regard, notably coagulation-precipitation by flocculation [4], membrane cake filtration [5], and electrocoagulation [6]. From these techniques, the adsorption process has been proven to be a very reliable and cost-effective process for the removal of pollutants [7].

Methylene blue is among the organic dyes present in the waste stream generated from the coloring industry processing textiles, paper, leather, food, and cosmetics [8]. In addition, methylene blue (MB) is a commonly used dye as a disinfectant in medicine, pharmaceuticals, pesticide production, varnish, and lacquer manufacturing [9]. It is a cationic dye with a chemical structure presented in Figure 1. It is known to cause various undesirable damage to living creatures, including permanent eye burns, breathing disorders, heart rate increases, tissue necrosis, nausea, vomiting, mental confusion, and painful micturition [10]. For this reason, the dye methylene blue was chosen as a cation dye model in this study. Among the conventional methods for removing methylene blue and other dyes from wastewater, adsorption is the most efficient. The adsorption method is the most widely used since it is simple, can be performed at low cost, and offers good efficiency. However, the available technological means to improve the adsorption process are adverse. Adsorption is a surface phenomenon based on the interaction between adsorbents in the solid phase and adsorbates in the liquid or gas phase [10,11,12,13]. The efficiency, capacity, and reusability of the adsorbent material depend on the functional groups on the adsorbent surface. In this case, the adsorption efficiency is controlled by the type and the strength of the interaction between the functional groups of the adsorbent and the adsorbate. So, the adsorbent selection is a crucial factor in this process [14]. Natural-based materials [15] and agricultural industrial by-products [16] are examples of low-cost adsorbents with functionalities that have a high affinity for MB.

Among these, clay is widely used as an adsorbent because it has advantages such as low cost, abundant availability, non-toxicity, and large surface area. Clay minerals that harden when dried are flexible by nature. These are fine-grained minerals that are naturally found on the soil surface [17,18]. Natural clay has limited applications due to its low adsorption capacity. Yet, its chemical surface yields the possibility of being modified and/or functionalized [19]. Marl, known as cement clay, is a low-cost clay composed of 50–70% limestone and 30–50% clay. In the cement industry, it is the only natural raw material where clay and limestone are found together.

In recent years, polymeric fibers have been used as adsorbents for the removal of dyes from aqueous media due to their relatively low cost, high specific surface area, and fast adsorption kinetics. For example, IDA-GMA-g-PET fibers have been exploited for the removal of the dyes Malachite Green (MG) and Rhodamine B (RB) [20]. MB has been removed using plasma activated acrylic acid grafted polypropylene (AA-PP) [21] and β-cyclodextrin graft modified polypropylene (PP) [22]. Graphene oxide (GO)-containing hydrogels were also investigated due to their high surface area and ability to adsorb a large amount of water and water-soluble molecules [23]. However, cellulose is the most abundant material in nature. Its derivatives have been widely used in water treatment [24]. Cellulosic fibers are considered low-cost biosorbents and ideal functionals substrates for the removal of organic and inorganic pollutants [25]. They can be easily modified by grafting various functional groups onto the surface-active hydroxyl moieties [26]. Moreover, modified cellulose could be prepared as a functional component of a hybrid composite by incorporation with other materials [27,28,29,30].

Activated carbon or material-based activated carbon was the most used adsorbent because of its high capacity to adsorb organic matter [23,31,32]. 

The use of Hap-based composites as adsorbents has raised great interest because of their efficiency and accessible cost [33,34,35,36,37].

Calcium phosphate materials, such as Haps, are unique inorganic compounds and the main mineral constituent of teeth, bones, and phosphate mineral materials. Because of its high capacity to remove divalent heavy metal ions, synthetic hydroxyapatite has been used to remove heavy metals such as Pb, Cr, Zn, Cu, Cd, Co, V, Ni, and Sb from water [38,39,40,41,42,43,44,45]. 

Recently, the synthesis of this type of phosphocalcic material has been widely studied in the literature for many applications [29,46,47]. In this context, the modification of the surface of hydroxyapatite to form composites with various organic and inorganic substances should give new functions to this material [48]. In fact, the interaction between hydroxyapatite calcium and biopolymers has been the subject of numerous studies, including HPMC, carboxymethylcellulose [49], collagen [47,48,49], polycaprolactone [50], sodium caseinate [51], Agar-Agar [52], chitosan [53], alginate [54], and gelatin [55,56]. These composite materials have been developed for a large variety of applications: bone substitution [57], tissue repair engineering [58], and biomedical and environmental fields [59].

The HPMC is a cellulose derivative in which part of the (OH) groups are replaced by a methyl and hydroxypropyl group Figure 2. These changes increased its solubility in an aqueous solution.

HPMC is an odorless and flavorless additive unmetabolized by humans, resistant to enzymes, and stable in pH between 2 and 13.

The objective of this study is to develop a new, efficient, and inexpensive adsorbent composite capable of interacting with the cationic dye MB. The dissolution/reprecipitation process was used for making the composite [61], which will allow the elaboration of nano-composites based on hydroxyapatite [Ca_10_(PO_4_)_6_(OH)_2_] and HPMC. Subsequently, several composites with various ratios of HAp/HPMC were synthetized. Then, the ability of HAp/HPMC composites to remove MB ions from aqueous solutions under various conditions was investigated.

## 2. Materials and Methods

### 2.1. Materials

The reagents and solvents used in this work were acquired from Sigma Aldrich (Burlington, MA, USA) and used without further purification. Reagents include: calcium nitrate tetrahydrate (Ca(NO_3_)_2_·4H_2_O, 99%), diammonium hydrogen phosphate ((NH_4_)_2_·HPO_4_, 99%), methylene blue C_16_H_18_ClN_3_S·H_2_O (molecular weight 319.85 g/mol), hydroxypropyl methylcellulose (molecular weight 86 kDa), ammonia solution NH_4_OH (Supelco), and perchloric acid HClO_4_, 70%.

### 2.2. Hydroxyapatite/Hydroxypropylmethylcellulose HAp/HPMC Composites

Hydroxypropyl methylcellulose is a hydrophilic cellulosic derivative soluble in cold aqueous solutions. The HAp/HPMC/H_2_O solution was prepared using the dissolution/reprecipitation process [62]. An amount of 1.5 g of hydroxypropyl methylcellulose was immersed into 150 mL of distilled water under continuous stirring at room temperature. This preparation was labeled as solution A. Hydroxyapatite solution was prepared by mixing 0.5 g of hydroxyapatite with 50 mL distilled water containing 0.5 mL HClO_4_ (*d* = 1.67 g/cm^3^) under continuous stirring at room temperature. The last solution was identified as solution B. Solution B was poured dropwise to solution A. The produced mixtures were brought to an adequate basic pH value by adding NH_4_OH. The mixtures were stirred for 2 h at 30 °C to obtain translucent colloidal suspensions. The translucent solutions obtained were poured into Petri dishes and left to dry at room temperature for 3 days.

### 2.3. Methods

In this study, the Infrared spectra were recorded using a Nicolet 6700 Fourier Transform Infrared (FT-IR) spectrometer supplied with the Smart SplitPea micro-ATR accessory (Shimadzu Scientific, Duisburg, Germany). The FT-IR spectra were collected at a resolution of 4 cm^−1^, covering the spectral range of 600–4000 cm^−1^ and 64 scans. 

The scanning electron microscopy (SEM) analysis was performed using a SU 8020, 3.0 KV SE (Hitachi High-Technologies Corporation, Tokyo, Japan). Before the analysis, all specimens were frozen using liquid nitrogen, fractured, mounted, coated with gold/palladium, and observed using an applied bias of 10 kV.

The standard thermogravimetric analysis was performed on the composites using the TGA Q500 and Q50 TA instrument (Shimadzu, Germany) at a temperature range from 20–900 °C and a heating rate of 10 °C/min. 

X-ray Diffraction (XRD) model (XPERT-PRO, PW3050/60) (LabxXRD-6100 Shimadzu, Germany) was used to analyze the powder diffraction at room temperature using a Diffractometer with CuKα radiation (1.5418 Å) in the range of 20° ≤ 2θ ≤ 80° using a sweep rate of 2°/min.

### 2.4. Surface Characterization

The surface analysis was performed using several instruments. The Atomic Force Microscopy (AFM) was used to study the cellulose blend surface morphology changes at 303 K. The measurements were performed utilizing a VEECO CPII atomic force microscope model (MPP-11123, Germany) at the cantilever’s resonance frequency between 335–363 kHz with a variable spring constant (*k*) of 20–80 N/m. During the X-ray photoelectron spectroscopy (XPS) experiment, the spectra were recorded using a Physical Electronic (PHI 5700 spectrometer, Germany) using a hemispherical multichannel detector with a constant pass energy mode at 30.1 eV, involving a 730 μm diameter analysis area. The data were analyzed using PHI ACESS ESCA-V6.0F software. In order to find the binding energy values, a carbon C_1s_ signal with a value of 284.6 eV was used for the characterization of the rest of the binding energies [63]. During the analysis, Shirley-type background and Gauss-Lorentz curves were involved.

### 2.5. Adsorption 

The cationic dye MB was selected for this study. The adsorption process was carried out using the batch method [64,65]. In this method, a 10.0 mg sample of the adsorbent was used per 10 mL of MB solution with various concentrations ranging from 40 to 300 mg/L. The adsorption was carried out at 25 °C at a stirring speed of 250 rpm. The effect of pH and the adsorption time on the adsorption efficiency were evaluated. The pH was adjusted by the addition of either HNO_3_ or NH_4_OH [65,66,67].

Samples of the reaction mixture were taken at regular time intervals to monitor the residual MB concentration. The samples taken were filtered and analyzed using a spectro UV–visible spectrophotometer type (MACY UV-1100, Germany) at a wavelength of 665 nm corresponding to the maximum MB adsorption (*λ*_max_). The residual concentration of the dye solution was calculated from the external standard curve.

The change in the concentration of MB was monitored by UV spectrophotometry. The nanocomposites’ adsorption capacity for MB was calculated as shown in Equations (1) and (2) [23]:(1)$$R\;\left(\%\right)=\frac{C_{0}-C_{e}}{C_{0}}\;100$$
(2)$$Q_{e}=\frac{C_{0}-C_{e}}{m}V$$

$C_{0}$ and $C_{e}$ are the initial and equilibrium concentrations (ppm) of cadmium in solution, respectively. $Q_{e}$ (ppm) is the equilibrium adsorption capacity, m is the weight of the adsorbent (mg), R the capacity for adsorption, and V is the volume of the solution (L).

### 2.6. Zero-Point Contaminated pH (pHpzc)

The zero-charge point, pHpzc, is the pH at which the average surface charge is zero [54], but this does not mean that there are no positive and negative charges; it means that the number of charges in the two categories are identical ([MO^−^] = [MOH_2_^+^]). This parameter is very important in adsorption phenomena, especially when electrostatic forces are involved in the mechanisms.

A quick and easy way to determine pHpzc is to use the pH drift method (potentiometric titrations). In this method, several flasks (100 mL), each containing a 60 mg sample of the composite and 20.0 mL of 0.1 M KNO_3_. The initial pH of each solution was adjusted in the range between 4 and 11 by adding a small volume of NaOH or HCl solutions (0.1 M) and measured on pH/ion meter. The suspensions were kept stirred at room temperature for 6 h, and the final pH was then determined. The pHpzc is the point at which the final pH versus initial pH curve intersects the pH_final_ = f(pH_initial_) regression line [68,69].

### 2.7. Adsorption Analysis

Langmuir (Equation (3)) and Freundlich isotherm (Equation (5)) models were applied to investigate the adsorption equilibrium between MB solution and the HAp/HPMC composite polymer [51]. Both models were used to assess the MB dispersion on the surface of the HAp/HPMC composites once equilibrium was reached under constant temperature.

The factor dictating the type of isotherm model is the correlation coefficient, R^2^ [51].
(3)$$\frac{C_{e}}{Q_{e}}=\frac{1}{q_{max}}C_{e}+\frac{1}{q_{max}K_{L}}$$
where $C_{e}$ represents the equilibrium concentration of the adsorbate (mg/L), $Q_{e}$ is the amount of the adsorbate adsorbed per unit mass of HAp/HPMC composite at equilibrium (mg/g), $q_{max}$ is the adsorption capacity equilibrium (mg/g), and $K_{L}$ is the Langmuir affinity constant (L/mg).

In the Langmuir isotherm model, the $R_{L}$ ratio is sometimes referred to as a dimensionless quantity indicating whether sorption is favorable or not.
(4)$$R_{L}=\frac{1}{1+K_{L\;}C_{0}}$$
where $C_{0}$ is the initial adsorbate concentration. If the value of $R_{L}$ is higher than 1, the adsorption is unfavorable. However, when the $R_{L}$ value is between 1 and 0, the adsorption is favorable. When $R_{L}=1$, adsorption is linear.

The Freundlich isotherm is an empirical formula used for low concentrations and is presented as [51]:(5)$$\;\ln(Q_{e\;})=\ln K_{F}+\frac{1}{n}\ln Ce$$
where $K_{F}$ is the Freundlich constant that deals with adsorption capacity (mg/g) and n is the heterogeneity coefficient which describes how favorable the adsorption process is (g/L).

## 3. Results

The current study presents a natural-based composite for the removal of methylene blue from wastewater. The composite was synthesized by dissolution/reprecipitation process using two natural-based materials, hydroxyapatite and hydroxypropyl methylcellulose. The composite was deposited as a membrane. The membrane showed excellent optical properties. Macroscopically, it is a thin and transparent film (Figure 3a,b). The film shows excellent dispersion of the apatite grains (inorganic phase) in the organic matrix.

The FT-IR spectra of HAp, HPMC and a composite of 30 wt.% HAp and 70 wt.% HPMC are overlaid in Figure 4. The spectrum of HPMC shows a band at about 3460 cm^−1^ corresponding to the stretching vibrations of the O-H bond. The Infrared spectrum also shows a band at 2800 cm^−1^ due to the stretching vibration of C-H bonds concerning the methyl and hydroxypropyl groups of HPMC. The band at 1643 cm^−1^ corresponds to the vibrations of the cyclic C-O bond [70]. The band at 1093 cm^−1^ corresponds to the elongation vibration of the glycosidic linkage C-O-C [70,71]. 

The FT-IR spectrum of HAp shows bands at 1090, 1033, 962, 603, and 561 cm^−1^, which could be attributed to the PO_4_^3−^ clusters of the apatite network. The composite shows a spectrum with different peaks that have different HAp, intensity, and wavenumber compared to those of the starting material, which indicates the presence of strong interactions between the composite components.

The HAp/HPMC composite was also subjected to analysis by solid-state ^13^C NMR to get a better picture of the strength of the interaction between the composite components compared to HPMC in Figure 5. The HPMC spectrum peaks are assigned as follows: C1′ (101–104 ppm), C2′-5′ (72–85 ppm), and C6′ (58–64 ppm). Those from HPMC were assigned to C1*(104 ppm), C2* (97 ppm), C3*-5*,2-6* (70–90 ppm), C6.c (67 ppm), and Ca (20 ppm). 

In the spectrum of HAp/HPMC grafted film, the resonance peak due to C-band Cb* (methoxy groups) of HPMC showed a shoulder at ∼59 ppm, which may be due to C6 of HPMC; we noticed a movement toward 57 ppm of HAp/HPMC composite.

In our study and analysis, the resonance peak, which is related to the methoxy carbons of HPMC, was labeled into three Voigt components, one at 59.30 ppm (Cb) and the other two at 61.42 ppm (Cb*). In the case of the spectrum of HAp/HPMC grafted composite, the identified Voigt component, which refers to C band Cb*, was shifted to 59.63 ppm and 61.52 ppm involving an increase in the peak area of the latter. This is probably due to the interference of the peaks corresponding to the C6 of grafted HAp/HPMC.

### 3.1. X-ray Diffraction

The prepared composites were characterized by X-ray diffraction. The XRD diffractograms of the HAp/HPMC composite, with weight ratios of 30 to 70, are shown in Figure 6. The composite shows that the apatitic structure is retained. The phenolphthalein test performed just after calcination of the composites at 900 °C showed the absence of the pink color (negative), which indicates a total absence of calcium oxide. The obtained Ca/P atomic ratio was less or equal to the stoichiometric hydroxyapatite (1.67) [72].

The width at mid-height of the index lines (002) and (310) was used to determine the average crystallite size using Scherrer’s formula [60,61,62,63,64,65]. The dimensions of the crystallites in the directions perpendicular to the (002) or (310) planes are given in Table 1. According to the obtained results, the dimensions of the nanoparticles evolve and are inversely proportional to the ratio of HPMC polymer to hydroxyapatite. So, the higher the ratio of HPMC polymer to mineral phase, the more space the apatite particles have to be deposited separately. The particle size was also calculated using Scherrer’s formula (6) shown below [73,74,75,76,77,78,79].
(6)$$D_{p}=\frac{0.94\;\lambda }{\beta \cos\theta }$$
where 0.94 stands for a geometrical factor that depends on crystallite apparent radius of gyration from the perspective of reflections with Bragg angle *θ* for X-rays of wavelength λ.

### 3.2. Thermal Analysis of HAp/HPMC Composite

The TGA curve of the HAp/HPMC (30/70) composite is shown in Figure 7. The curve shows two mass-loss events. The first is recorded between 50 and 100 °C, which corresponds to the loss of water. The second mass loss between 200 and 350 °C is attributed to the decomposition of the organic matter represented by HPMC. The two mass losses were confirmed by DSC as two endothermic peaks at 98 °C and 350 °C. The DSC reveals an endothermic transformation around 940 °C due to the partial decomposition of non-stoichiometric hydroxyapatite.

### 3.3. Scanning Electron Microscopy

Scanning electron microscopy images of the HAp/HPMC composite (70/30) are shown in Figure 8. The images show a homogenous dispersion of hydroxyapatite particles in the polymer matrix with a nano size. The surface of the sample was irregular in shape and exhibited a rough morphology and agglomerated surface or in the form of a nano to micro-sized lump.

### 3.4. Chemical Composition of the Composite

The main chemical elements of HAp/HPMC and their percentages are summarized in Table 2. 

XPS analysis was carried out on the HAp/HPMC composite (70/30) in order to obtain information about the Ca/P ration. This technique would be a good way to assess which termination (rich in calcium or rich in phosphate) is closest to the reality of the composite. It will simply be specified that the peaks used to obtain the Ca/P ratios are those relating to the 2p orbitals of calcium and phosphorus. Figure 9 shows that in favorable conditions for preparing the composite based on hydroxyapatite, the Ca/P ratio is lower than those obtained by chemical analysis; the surface is, therefore, less rich in calcium than the whole sample. This phenomenon was also observed by Tanaka, who deduced that the Ca atoms on the surface area were covered by an excess of phosphate ions [77]. It is possible to conclude that this difference between core and surface composition is quite specific to HAp.

### 3.5. Adsorption of Methylene Blue on the Synthesized HAp/HPMC Composites

In this work, we are interested in using the prepared HAp/HPMC composite as an adsorbent of methylene blue colorant and determining the optimum adsorption conditions and composite component ratios.

### 3.6. Determination of the Zero-Load Point of HAp/HPMC Composite

The potentiometric titration method was performed in order to consider if BM at the surface of HAp/HPMC composite changed adsorbent surface charge by determination of the point of zero charge. The evolution of pH as a function of initial pH is shown in Figure 10. The first part of the curve, between the initial pH at 4 or 5, shows that the final pH of the solution increases as the initial pH increases. This is explained by the fact that the H^+^ ions introduced into the solution are consumed by the surface of the material. Between initial pH 6 and initial pH 8 the curve becomes parallel to the *x*-axis (i.e., the final pH of the solution is stable). Indeed, all H^+^ ions introduced in the solution are consumed by the surface of the material until saturation of the sites; then, there is a decrease in the final pH due to the consumption of OH^−^ ions via the deprotonation of the existing sites on the surface of the composite material.

When the pH value is higher than pH_pzc_, the surface of the solid is negatively contaminated, which favors the adsorption of cationic species. Meanwhile, when the pH values < pH_pzc_, the surface is positively contaminated, favoring the adsorption of anionic species. The pH_pzc_ of the HAp/HPMC (60/40) composite is 6.4.

The results obtained show that the contamination developed on the surface of the composite material depends on the pH value.

Figure 11 shows the variation of the surface charge of HAp/HPMC composite as a function of pH.

The pH has a very important role in the adsorption phenomenon, as it can influence both the structure of the adsorbent and the adsorbate; it also might affect the adsorption mechanism. The study of the influence of pH on the adsorption of MB by the HAp/HPMC composite powder was performed using an initial MB concentration of 53.5 mg/L and a pH ranging from 6 to 12. The evolution of the amount of MB adsorbed as a function of pH is shown in Figure 12. The other variables, such as temperature, time, adsorbate mass, and the mixing speed, were kept constant at 25 °C, 30 min, 20 mg, and 250 rpm, respectively. 

The adsorption behavior of methylene blue on adsorbents has been studied over a wide pH range from 6 to 12. Adsorption results of MB on the composite show that the maximum adsorption was observed at pH = 8. The increase in the pH value from 6 to 12 was accompanied by a slight increase in the amount of dye adsorbed by HAp/HPMC composites. This behavior may be due to the fact that the surface of the HAp/HPMC composite is negatively contaminated at pH values above pH_pzc_, which promotes the adsorption of the cationic dye (methylene blue). On the other hand, for pH values below pH_pzc_, the surface of the HAp/HPMC composite is positively contaminated and, therefore, likely to repel the colorant (the pH_pzc_ of HAp/HPMC is 6.4). As the pH decreases, the number of negatively contaminated sites decreases and the number of positively contaminated sites increases. When the percentage of HPMC increases in the composite, the adsorption increases, indicating an increase in the number of preferential sites for MB adsorption.

By definition, adsorption kinetics represent the progress of the adsorption process of the dye on the adsorbent surface as a function of contact time. It is a very important criterion to consider when evaluating the performance of the selected adsorbent. A good adsorbent must have not only a good adsorption capacity but also a good adsorption rate. The adsorption kinetics of the MB colorant on composites (HAp/HPMC: 80/20, 70/30 and 60/40) were studied by determining the adsorption rate as a function of time while the other variables were kept constant (temperature 25 °C, pH 8.0, adsorbate mass 53.5 mg (20.0 mL), adsorbent 20 mg, and a mixing speed of 250 rpm). Sample analyses were taken at regular time intervals to determine the residual concentrations of the MB colorants. The obtained results are shown in Figure 13.

The results obtained reveal that the quantity of MB bound to the composite increases with time and reaches a maximum adsorption of approximately 51.78 mg/g at 20 min for HAp/HPMC 60/40.

The adsorption at the beginning was rapidly related to the presence of a large number of vacant sites on the surface of the composite used, whereas the slow rate is explained by the decrease in the number of vacant sites and the difficulty of accessing them due to the repulsion forces between MB and the liquid phase.

The effect of initial MB dye concentrations on the adsorption efficiency was evaluated using various initial concentrations ranging from 5 mg/L to 90 mg/L. The adsorption runs were performed at a pH value of 8. The other variables were kept constant, as shown above. 

Results are summarized in Figure 14; the results show that the adsorption capacity of the composites increases with the increase in the initial MB concentration, then becomes constant at 40 mg/L. During the adsorption process, MB molecules first reach the surface layer and physically bond to the surface-active sites. Once the surface sites become saturated, the MB molecules diffuse through the surface layer into the porous part of the inner structure of the sorbent. This explains the increase in the rate of adsorption with time. 

### 3.7. Adsorption Isotherms

Adsorption isotherms are commonly used to describe a relationship between the concentration in an aqueous solution and the amount fixed on the adsorbent when the adsorption process reaches equilibrium. The adsorption isotherms were established by contacting aqueous solutions of MB with concentrations ranging between 5 mg/L and 90 mg/L, and 10.0 mg composite, at a pH value of 8 and at room temperature. The Langmuir (Equation (3)) and Freundlich models (Equation (4)) for the adsorption of MB by the composites were examined. Results are summarized in Figure 15 and Figure 16.

Adsorbent and adsorbate contact time was kept for 2 h under agitation after analysis and determination of the residual concentration. The linear representations of the obtained experimental values of this adsorption process were used to determine the equilibrium parameters and the values of the Langmuir and Freundlich constants calculated by linear regression Table 3 and Table 4. The values of the regression coefficients indicate that the adsorption process of MB by the composites follows the Langmuir isotherm (with excellent linear regression coefficients *R*^2^, which are very close to one). This implies that the adsorption of MB on the composites occurred in the form of monolayers. The values of *Q*_max_ and *K*_l_ were obtained from the intercept *C*_e_/*Q*_a_ = *f*(*C*_e_). The maximum adsorption capacities are 59.52 mg·g^−1^ for HAp/HPMC 80/20 and 59.88 mg·g^−1^ for HAp/HPMC 60/40.

A summary of the adsorption efficiency of various adsorbents toward MB is summarized in Table 5.

The values are collected from the published literature. As shown in Table 5, the adsorption efficiency of the 60/40 composite is as good as the ordered mesoporous silica, which shows the highest efficiency [87].

### 3.8. Adsorption Kinetics

Adsorption kinetics allow estimating the quantity of pollutant adsorbed as a function of time. The kinetics also provide information on the adsorption mechanism and the mode of transfer of the solute from the liquid phase to the solid phase.

Two models have been applied to describe the mechanism of adsorption kinetics, the pseudo-first-order and the pseudo-second-order. Figure 17 shows that ln(*Q*_e_ − *Q*_t_) as a function of time is non-linear. It is deduced that the adsorption kinetics of Methylene Blue on HAp/HPMC cannot be described by pseudo-first-order kinetics. In addition, *R*^2^ values were found to be relatively low. The calculation of *Q*_e_ for the different percentages shows that the adsorbed amounts of the MB are rather small compared to the experimental amounts. On the other hand, when comparing the linear representation Figure 18 of *t*/*Q*_e_ as a function of time, the adsorption capacities at equilibrium and the correlation coefficients calculated for the pseudo-second-order (Table 6) show that this model can describe the kinetic behavior of methylene blue adsorption on the HAp/HPMC composite. Indeed, it was noticed that the correlation coefficients *R*^2^ (pseudo-second-order) are very close to 1, and the values of the adsorption capacities calculated (*Q*_e cal_) from the pseudo-second-order model were very close to the values obtained experimentally (*Q*_e exp_).

## 4. Conclusions

In this work, natural-based nanocomposites in the form of membranes were synthesized from HAp and HPMC using the dissolution/re-precipitation method to elaborate nanocomposites. The synthesized composites were characterized by scanning electron microscopy (SEM), calorimetry/thermogravimetric analysis (DSC/TGA), infrared spectroscopy (FT-IR), X-ray photoelectron spectroscopy (XPS), ^13^C CP/MAS NMR, X-ray fluorescence spectrometry (XRF), and X-ray diffraction (XRD). The composites, as shown by the obtained analysis results, showed a uniform blend with strong interactions between their components. The HAp/HPMC composites were evaluated as adsorbents for the dye synthetic MB form in an aqueous solution. The experimental results showed that the adsorption process can be controlled by various parameters such as solution pH, initial concentration of the dye, and time. The amount of dye adsorbed showed an increase as the pH value increased. It was determined that the zeta potential of the clay increased negatively as the pH increased.

The adsorption kinetics of MB on HAp/HPMC composites shows that the adsorption process is very fast, and equilibrium is established after 20 min. The adsorption mechanism was effectively described by pseudo-second-order kinetics.

Langmuir’s model adequately describes the adsorption isotherms of MB on HAp/HPMC composites. The maximum adsorption capacities determined according to the Langmuir isotherm of the composites (HAp/HPMC: 80/20, 60/40) are respectively 59.52 and 59.88 mg·g^−1^. These values indicate that the HAp/HPMC 60/40 composite exhibited the best adsorption capacity.

The results of the experiments showed that the dye removal efficiency increases with increasing initial dye concentration.

As a result, it is seen that the removal of cationic dyes, which are an important problem in industrial water treatment, can be achieved by using apatitic composite biomaterials.

## Figures and Tables

**Figure 1 polymers-14-02147-f001:**
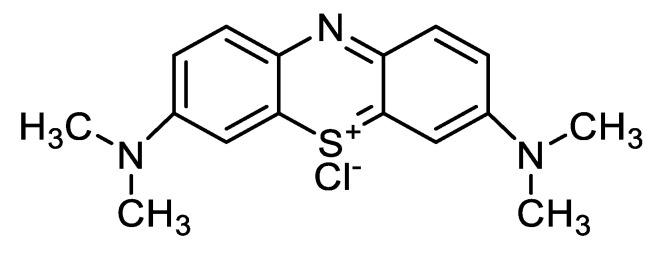
Chemical structure of methylene blue C_16_H_18_ClN_3_S × H_2_O.

**Figure 2 polymers-14-02147-f002:**
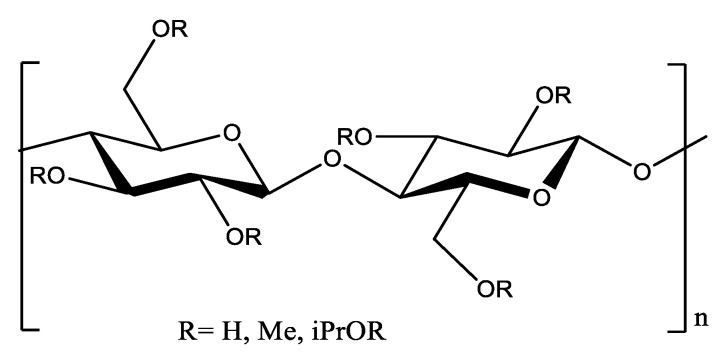
Structure of (HPMC) [60].

**Figure 3 polymers-14-02147-f003:**
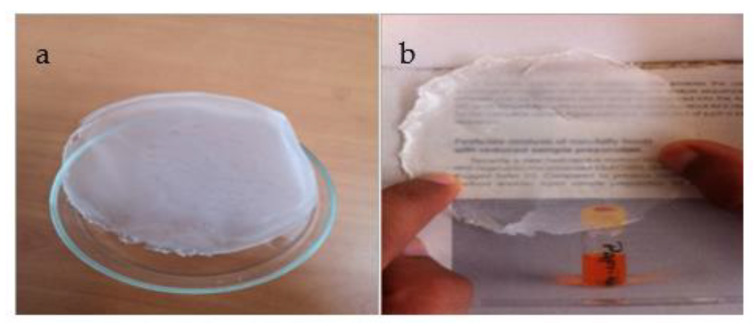
Image of the HAp/HPMC membrane. (**a**) thin and homogenous film and (**b**) transparent film.

**Figure 4 polymers-14-02147-f004:**
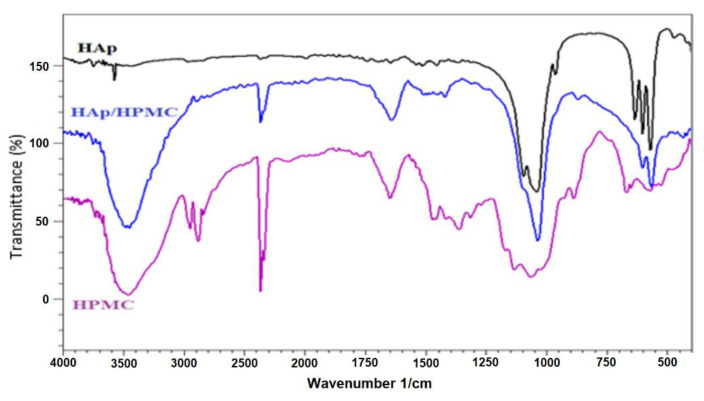
FT-IR spectra of HAp alone, HAp/HPMC and HPMC.

**Figure 5 polymers-14-02147-f005:**
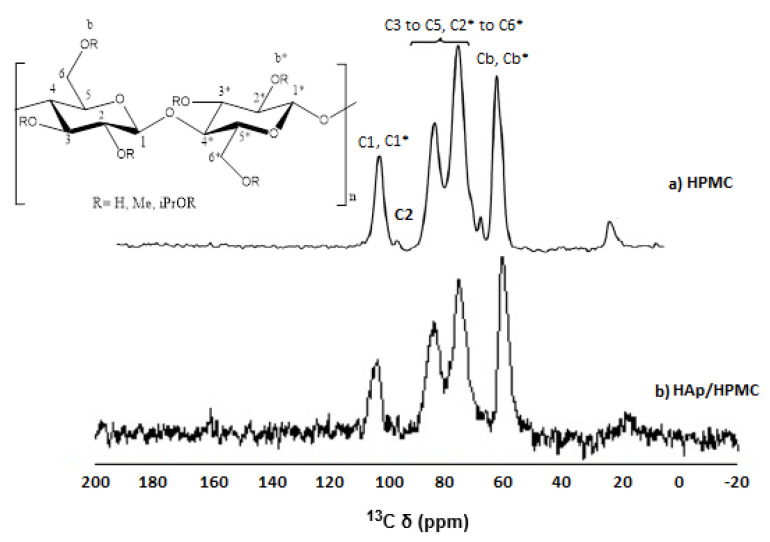
^13^C CP-MAS solid- state NMR spectra (**a**) HPMC (**b**) HAp/HPMC.

**Figure 6 polymers-14-02147-f006:**
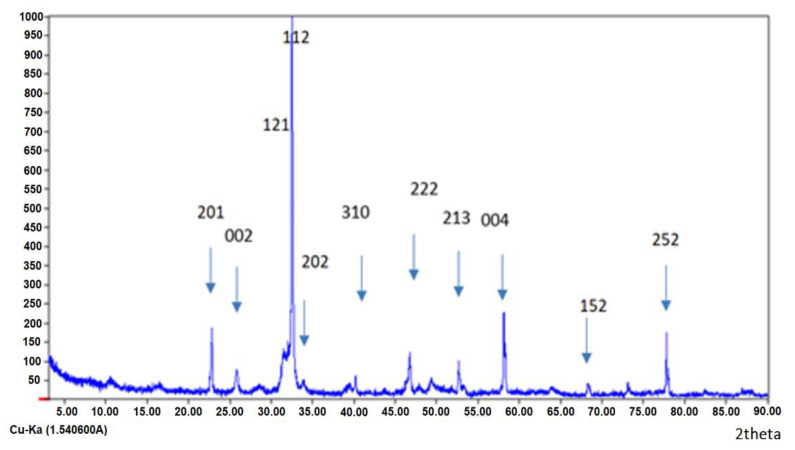
XRD diffractogram of the HAp/HPMC 30/70 composite.

**Figure 7 polymers-14-02147-f007:**
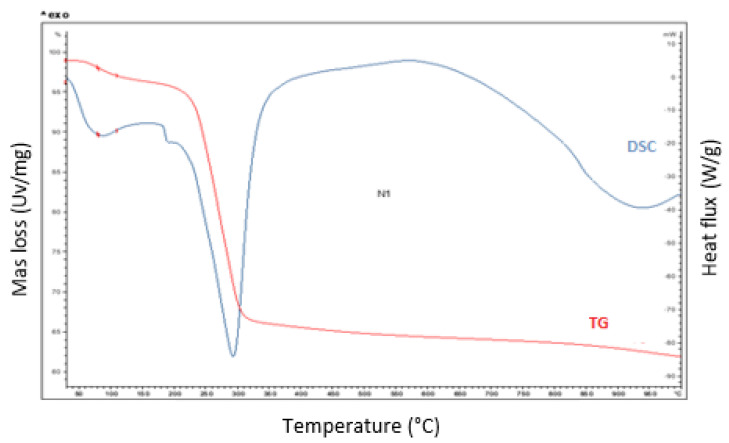
DSC/TGA curves of the HAp/HPMC composite.

**Figure 8 polymers-14-02147-f008:**
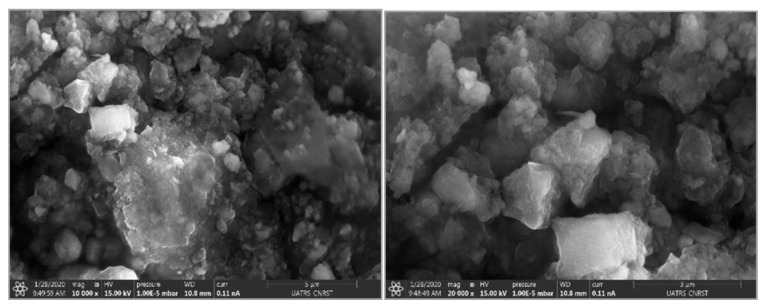
Scanning electron microscopy images of the HAp/HPMC composite (70/30).

**Figure 9 polymers-14-02147-f009:**
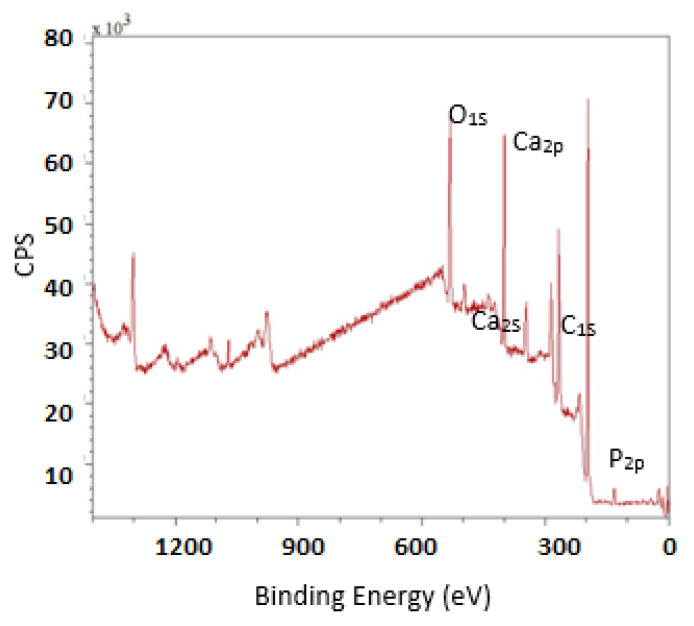
XPS spectra of the composite based on HAp/HPMC.

**Figure 10 polymers-14-02147-f010:**
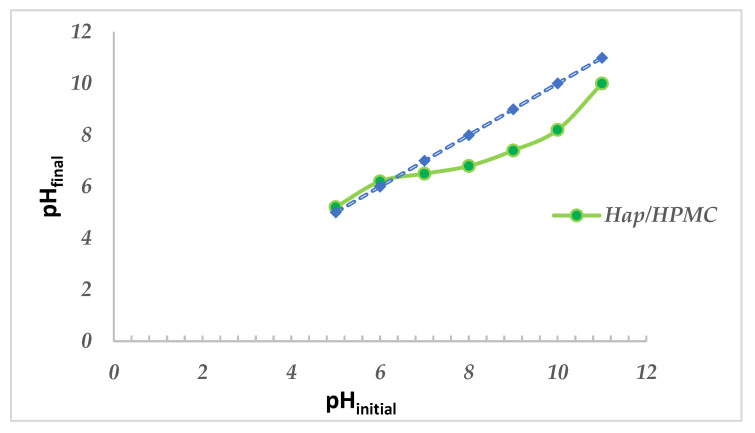
Determination of the pH_pzc_ of the composite material HAp/HPMC 60/40.

**Figure 11 polymers-14-02147-f011:**
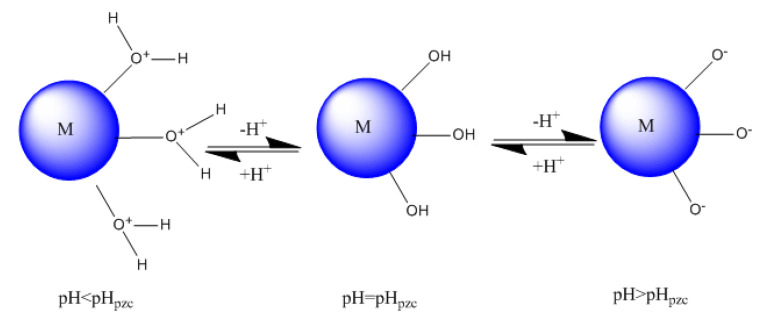
Change in surface contamination of the composite material (M represents HAp/HPMC composite) as a function of pH.

**Figure 12 polymers-14-02147-f012:**
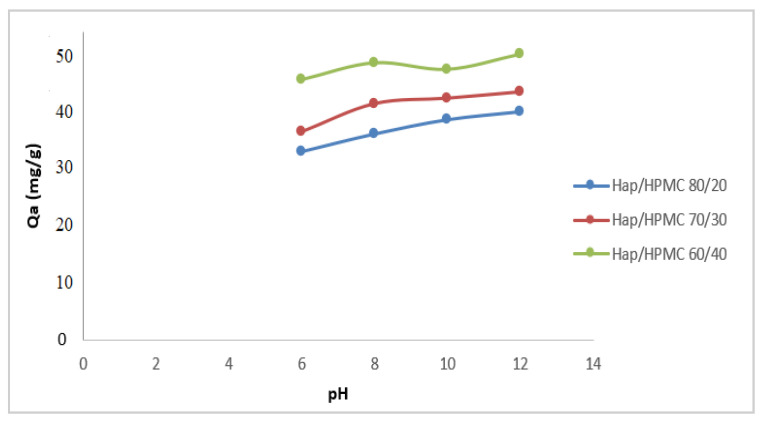
Effect of pH on the adsorption efficacy of the composites, [MB] = 53.5 mg/L and *T* = 25 ± 2 °C and *V*_rot_ = 250 rpm.

**Figure 13 polymers-14-02147-f013:**
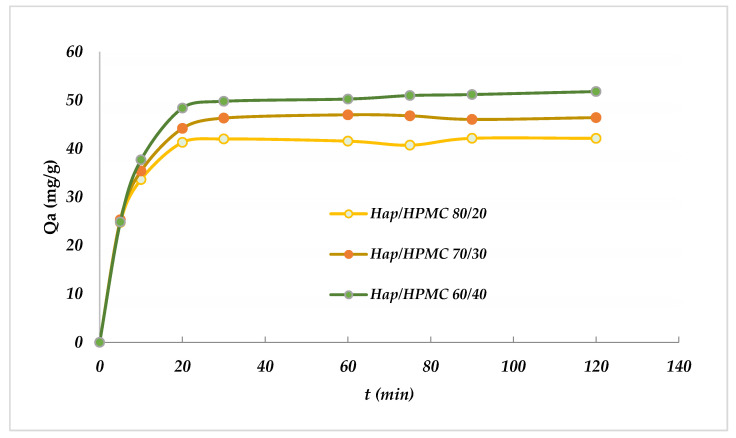
Evolution of the MB adsorbed quantity (*Q*_a_) as a function of time, [MB] = 53.4 mg/L, pH = 7 and *T* = 25 ± 2 °C, adsorbent 10 mg, *V*_rot_ = 250 rpm.

**Figure 14 polymers-14-02147-f014:**
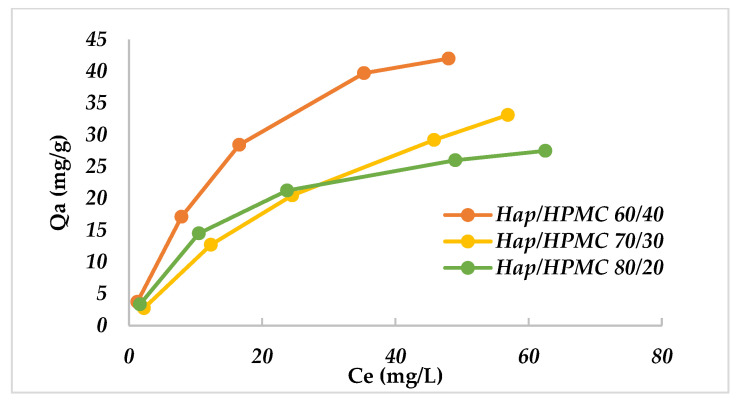
Evolution of the adsorbed quantity as a function of the MB concentration, pH = 7, *T* = 25 ± 2 °C and *V*_rot_ = 250 rpm.

**Figure 15 polymers-14-02147-f015:**
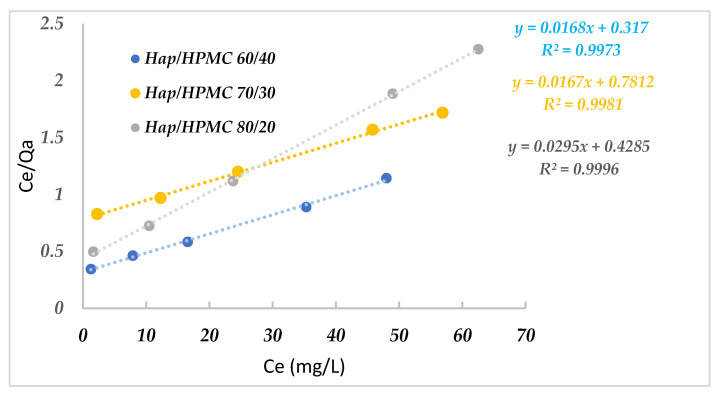
Linearization of the Langmuir equation for the studied adsorbent/adsorbate systems. *C*_e_: concentration of adsorbate, *Q*_a_: adsorbed quantity.

**Figure 16 polymers-14-02147-f016:**
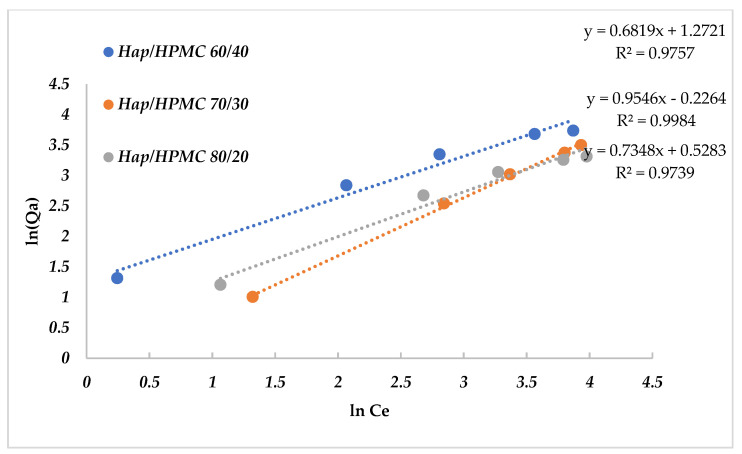
Linearization of the Freundlich equation for adsorbent/adsorbate systems. *C*_e_: concentration of adsorbate, *Q*_a_: adsorbed quantity.

**Figure 17 polymers-14-02147-f017:**
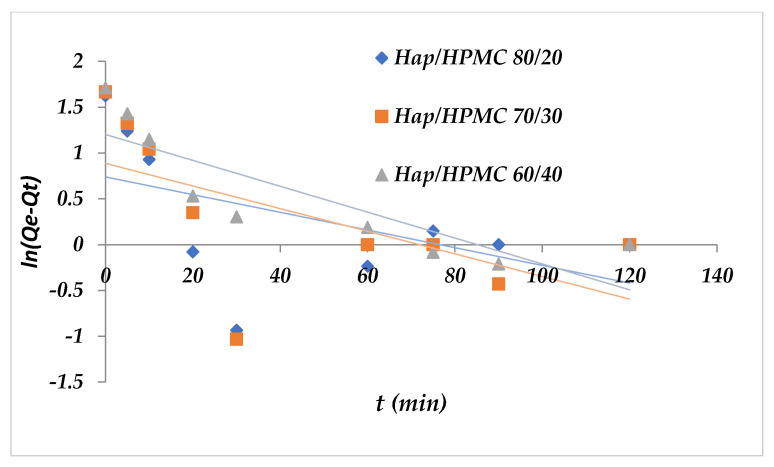
Pseudo-first-order kinetic model applied to the adsorption of MB by the composites.

**Figure 18 polymers-14-02147-f018:**
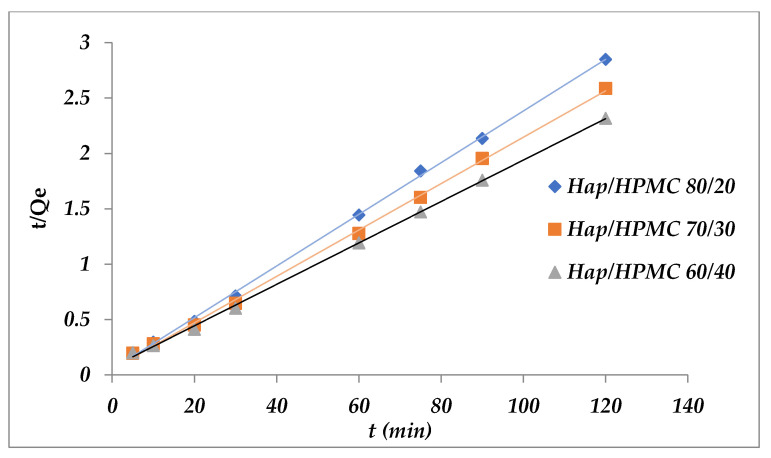
Pseudo-second-order kinetic model applied to the adsorption of MB by the composites.

**Table 1 polymers-14-02147-t001:** Average size of apatitic nanoparticles according to Scherrer’s formula.

Compounds	λ(Å)	Planes hkl	D (nm)	D_mean_. (nm)
Pure HAp	1.5406	002	63	63
		310	63	
HAp/HPMC 30/70	1.5406	002	50	47
		310	45	
HAp/HPMC 20/80	1.5406	002	23	23
		310	23	

**Table 2 polymers-14-02147-t002:** Contents of the main elements in the chemical composition of the raw materials used in this study (by XRF method).

Element	wt (%)	Est.Error
Ca	28.82	0.23
Cl	13.47	0.17
P	16.83	0.16

**Table 3 polymers-14-02147-t003:** Results using the Langmuir model of the MB adsorption isotherm on the HAp/HPMC composite.

Composites	*Q* _max_	*K* _L_	*R* ^2^
HAp/HPMC 60/40	59.88	3.89 × 10^−2^	0.9996
HAp/HPMC 70/30	53.9	6.88 × 10^−2^	0.9981
HAp/HPMC 80/20	59.52	3.920 × 10^−2^	0.9973

**Table 4 polymers-14-02147-t004:** Results using the Freundlich model of the MB adsorption isotherm on the HAp/HPMC composite.

Composites	*K* _F_	*n*	*R* ^2^
HAp/HPMC 60/40	3.568	1.466	0.9757
HAp/HPMC 70/30	0.797	1.048	0.9984
HAp/HPMC 80/20	1.327	1.361	0.9739

**Table 5 polymers-14-02147-t005:** Comparison of adsorption capacities of various adsorbents for MB removal.

Adsorbent	*Q*_m_ (mg/g)	pH	Temperature (K)	References
Fly ash	3.07	7.5	303	[80]
Posidonia oceanica	5.56	10	303	[81]
Ulva lactuca	10.99	10	298	[82]
Neem leaf powder	8.76	-	300	[83]
Natural phosphate	7.23	-	298	[84]
Coir pith carbon	5.87	6.9	308	[85]
Clay	6.3	-	293	[86]
Ordered mesoporous silica	54.0	-	295	[87]
Poorly crystalline HAp	14.27	9.0	283	[88]
Natural zeolite	16.37	-	-	[89]
Kaolinite	13.99	-	300	[90]
Sawdust	11.8	7.0	299	[91]
Phosphoric acid modified E	31.15	-	-	[92]
α-chitin nanoparticles	6.90	-	-	[93]
Activated carbon	47.62	-	-	[94]

**Table 6 polymers-14-02147-t006:** Kinetic parameters of MB adsorption on composites.

Composites	Pseudo-First-Order				Pseudo-Second-Order		
	*Q* _e_	*K* _1_	*R* ^2^	*Q* _e cal_	*Q* _e exp_	*K* _2_	*R* ^2^
HAp/HPMC 60/40	3.32	0.014	0.728	53.47	51.78	0.005	0.9992
HAp/HPMC 70/30	2.43	0.012	0.365	47.85	46.40	0.008	0.9989
HAp/HPMC 80/20	2.095	0.010	0.260	42.92	42.12	0.011	0.9991

## Data Availability

The data used to support the findings of this study are available from the corresponding author upon request.

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
