# Peer review of "Environmental-Friendly Adsorbent Composite Based on Hydroxyapatite/Hydroxypropyl Methyl-Cellulose for Removal of Cationic Dyes from an Aqueous Solution"

_polymers, 2022, doi:10.3390/polym14112147_

Round 1

Reviewer 1 Report

  • It is suggested to add a paragraph to the introduction section about the modified polymeric fiber-based adsorbents used for the removal of cationic dyes. The following references are recommended for citation:
    1. https://doi.org/10.1016/j.surfin.2021.101278
    2. https://doi.org/10.1080/19443994.2013.873350
    3. https://doi.org/10.1080/25740881.2019.1669650
    4. https://doi.org/10.1177/1558925019828194
  • Section 2.2., the preparation of the adsorbent should be described clearly.
  • Section 2.5, reference is needed.
  • Page 4, line 184; “Cu2+ and Zn2+ ion solution”?? line 185; metal ion?? do you mean MB solution?
  • How it was assumed from fig. 10 that the pHpzc is 6.4? the method is not clearly described.
  • Generally, the methods of the experiments and preparation of the composites are not well described. It needs careful revision.

Author Response

Dear reviewer;

thank you very much for your suggestions. please, find in attached file our reponses.

Best regards

Reviewer 2 Report

The manuscript under consideration describes nanocomposite based on hydroxyapatite (HAp) and hydroxypropyl methylcellulose (HPMC). Good number of experiments is provided. Material that reported in manuscript was probed with different techniques to show physical and chemical interactions. Also kinetics of the adsorption process was characterized.

Author Response

(The authors gave the same response as above.)

Reviewer 3 Report

"Environmental-friendly adsorbent composite based on hydroxyapatite/ hydroxypropyl methyl-cellulose for removal of cationic dyes from an aqueous solution" is written on a work of adsorbent for the cationic dye methylene blue. The authors have described work with a good degree of detail. The manuscript is in general good; however, the formal part is now unacceptable.

Overall the introduction is missing a more detailed explanation of the state of the art and within that explanation a deeper understanding of the adsorbent for wastewater treatment. The reviewer thinks that the authors should focus on this in the Introduction. There are many references, such as Adv Compos Hybrid Mater 3, 187–193 (2020). https://doi.org/10.1007/s42114-020-00146-4;  Adv Compos Hybrid Mater 4, 1384–1397 (2021). https://doi.org/10.1007/s42114-021-00357-3; Adv Compos Hybrid Mater 4, 65–73 (2021). https://doi.org/10.1007/s42114-021-00205-4..  

Some Figures are unnecessary, e.g., Figures 1-3. There are not important informations. In addition, most of the Figures are not clear.

Author Response

(The authors gave the same response as above.)

Round 2

Reviewer 1 Report

Accept

Reviewer 3 Report

accept

This manuscript is a resubmission of an earlier submission. The following is a list of the peer review reports and author responses from that submission.